# Serjanic Acid Improves Immunometabolic Markers in a Diet-Induced Obesity Mouse Model

**DOI:** 10.3390/molecules25071486

**Published:** 2020-03-25

**Authors:** Gustavo Gutiérrez, Deisy Giraldo-Dávila, Marianny Y. Combariza, Ulrike Holzgrabe, Jorge Humberto Tabares-Guevara, José Robinson Ramírez-Pineda, Sergio Acín, Diana Lorena Muñoz, Guillermo Montoya, Norman Balcazar

**Affiliations:** 1Natural Sciences School, Pharmaceutical Sciences Department, Universidad Icesi, 760031 Cali, Colombia; ggutierrezg94@gmail.com; 2School of Chemistry, Industrial University of Santander, 680003 Bucaramanga, Santander, Colombia; deisy.giraldo.davila@gmail.com (D.G.-D.); marianny@uis.edu.co (M.Y.C.); 3University of Würzburg, Institute for Pharmacy and Food Chemistry, 97074 Würzburg, Germany; ulrike.holzgrabe@uni-wuerzburg.de; 4Grupo Inmunomodulación, School of Medicine, Universidad de Antioquia, 050010 Medellín, Antioquia, Colombia; jorgetabare@gmail.com (J.H.T.-G.); jrobinson.ramirez@udea.edu.co (J.R.R.-P.); 5Department of Physiology and Biochemistry, School of Medicine, Universidad de Antioquia, Carrera 51D Nº 62–29, 050010 Medellin, Colombia; sergio.acin@udea.edu.co; 6GENMOL Group. Sede de Investigación Universitaria, Universidad de Antioquia, Calle 62 # 52–59, 050010 Medellín, Colombia; dhyana5@yahoo.com

**Keywords:** *Cecropia telenitida*, serjanic acid, type 2 diabetes

## Abstract

Plant extracts from *Cecropia* genus have been used by Latin-American traditional medicine to treat metabolic disorders and diabetes. Previous reports have shown that roots of *Cecropia telenitida* that contains serjanic acid as one of the most prominent and representative pentacyclic triterpenes. The study aimed to isolate serjanic acid and evaluate its effect in a prediabetic murine model by oral administration. A semi-pilot scale extraction was established and serjanic acid purification was followed using direct MALDI-TOF analysis. A diet induced obesity mouse model was used to determine the impact of serjanic acid over selected immunometabolic markers. Mice treated with serjanic acid showed decreased levels of cholesterol and triacylglycerols, increased blood insulin levels, decreased fasting blood glucose and improved glucose tolerance, and insulin sensitivity. At transcriptional level, the reduction of inflammation markers related to adipocyte differentiation is reported.

## 1. Introduction

Type 2 diabetes mellitus (T2DM) is a growing health issue worldwide. Lifestyles with high-calorie foods consumption and lower levels of physical activity have increased and will increase the prevalence of this disease. Diabetes is predicted to become the fifth leading cause of death in the world by the year 2030 in lower-middle-income countries according to global health estimates of the World Health Organization [1]. Obesity is the most critical metabolic disorder in establishing insulin resistance and the potential development of T2DM. Between 60% and 90% of reported cases of T2DM are related to obesity [1].

Obesity is a chronic condition that contributes to the development of a number of conditions, including cardiovascular disease (CVD), T2DM, and cancer [2]. In obesity induced by hyperphagia, as caloric intake increases, hypertrophy occurs in adipocytes due to the increase of stored triacylglycerols (TAGs). As hypertrophy reaches the upper limit and is maintained over time, the endocrine function of the visceral adipose tissue is altered. Hypertrophic adipocytes secrete several adipokines that are directly associated with inflammation and a microenvironment is established that induces oxidative stress, inflammation, and the release of nonesterified free fatty acids; the combination of these phenomena is involved in the generation of insulin resistance, in both adipose tissue and peripheral organs. Insulin resistance occurs when the effects of insulin are lower than expected, both in the uptake of glucose by skeletal muscle and adipose tissue and of endogenous glucose production suppression by the liver [3,4].

*Cecropia* (Urticaceae) is a neotropical genus frequently found in humid areas up to 2600 m above sea level. Species of *Cecropia* occur from Mexico to north of Argentina. The principal ethnopharmacological usage of Cecropia genus comprises anti-inflammatory properties, and the treatment of type 2 diabetes [5,6,7]. In contrast, the distribution of *Cecropia telenitida* [8], retrieved from the TROPICOS database (Missouri Botanical Garden), shows how this species is limited to the central and northern sections of the Andes (Venezuela, Colombia, Ecuador, and north of Peru). Despite the fact that *C. telenitida* has not been abundantly studied in terms of its chemistry and bioactivity, the genus has some phytochemical descriptions [7,9,10,11,12], and our research group has reported pentacyclic triterpenes (PT) as chemotaxonomic markers in the roots of *Cecropia* genus.

PT are secondary plant metabolites, which are widely distributed in the plant kingdom. These are found in variable amounts in edible vegetables and fruits making them regular constituents in the human diet [13,14,15,16]. Consumption of these naturally occurring substances seems to be associated with the prevention of metabolic syndrome and T2DM [17,18,19,20,21,22]. In vitro and animal studies have demonstrated the potential of PTs to modulate diseases ranging from diabetes to cardiovascular conditions [23,24,25,26,27,28]. Interestingly, most of the PT studied so far, exhibit anti-inflammatory, antioxidant, anti-obesity, and antidiabetic effects. The PT scaffold and their derives have been reported to have a variety of mechanisms of action, such as reduction of glucose absorption, endogenous glucose production inhibition, insulin sensitivity enhancement, lipid homeostasis improvement, and body weight regulation [22,29].

Serjanic acid (SA) is a PT identified in nature making part of complex glycosides mixtures from *Chenopodium quinoa* and *Phytolacca* genus without a detailed description of its potential properties [30,31]. SA is a noncommercialized molecule also present in *Cecropia telenitida* roots, and a standard protocol was developed to obtain a sufficient amount to support animal assays [10].

Our research assessed the favorable effect of SA modulating direct markers in obese insulin-resistant murine model such as glucose and insulin levels, lipidic markers (low density lipoprotein (LDL), high density lipoprotein (HDL), triglycerides (TAG), and total cholesterol), and gene expression of important adipokines (acetyl-CoA carboxylase (ACC), peroxisomal proliferation-activated receptor-alpha (PPAR-α), peroxisome proliferator-activated receptor gamma coactivator-1 (PGC-1), carnitine palmitoyltransferase 1-A (Cpt1A), tumor necrosis factor-alpha (TNF-α), monocyte chemoattractant protein-1 (MCP-1), interleukin-6 (IL-6), and interleukin-1-beta (IL-1β)).

## 2. Results and Discussion

### 2.1. SA Extraction and Purification

Despite substantial developments in extraction and purification techniques, natural products isolation is still a challenging task, and reproducibility has remained as the key concern when thinking in scale-up [32]. Our group has reported the presence of SA at the roots of *Cecropia telenitida* and was described as one of the most abundant PTs [12]. The laboratory has kept the SA standard previously isolated due to being not commercially available, and organic molecules isolation becomes easier having a reference material. Extraction protocol for PTs dealing with the same vegetal material was reported using a semi-pilot extraction plant coupled with and automated flash and low-pressure chromatography systems [10]. A MALDI-TOF support the choice of extracts having the major amount of SA, as shown in Figure 1, since thin layer chromatography of crude extracts has made us take wrong decisions. Thereafter, reference SA material and TLC of purified fractions conducted the rest of the purification stage.

From 5 kg *Cecropia telenitida* roots, 5 g of pure SA giving an extraction yield of 0.1% were obtained. The spectroscopic analysis was compared with reference material and the high-performance liquid chromatography-photodiode array (HPLC-PDA) method were carried out to define optical purity of more than 98%. To guarantee elimination of residual solvents of the molecule, a freeze-drying process removes traces of solvents used during purification.

### 2.2. Biological Activity Assessment

The present study assesses the effect of SA in the metabolic alterations evidenced in a murine model of diet-induced obesity, which is characterized by the development of a systemic low-grade inflammatory state with concomitant insulin resistance. Following the administration of 10 oral doses of metformin or SA, the animals on a high fat diet (HFD) showed decreased glucose levels during fasting and improved tolerance to glucose and insulin (Figure 2A–D). Decreased insulin levels during fasting were observed in animals treated with metformin and significantly increased levels were observed in those treated with SA (Figure 2E–H). In this context, it can be assumed that, although metformin administration reduces insulin resistance, SA does not affect it; however, it reduces glycemia, possibly increasing insulin levels in the blood. These results suggest that SA has a beneficial effect on pancreatic β-cells, thereby stimulating an increase in insulin secretion. This is supported by the values calculated to determine both insulin resistance (HOMA-IR) and pancreatic β-cell functionality (HOMA—%B) in the diet-induced obesity model used (Figure 2E,F) [33].

Fasting blood insulin was increased after 10 treatment doses of SA, this effect could be explained through different cellular mechanisms [34]. Wistar rats showed increased insulin level after oleanolic acid (OA) administration, and OA is a biosynthetic precursor of SA and therefore sharing high structural likenesses. The inhibition of cholrine uptake and acetylcholine transport employing hemicholinium-3 and vesamicol suppressed the OA positive effect, suggesting that OA promotes the acetylcholine release, which in turn, stimulates the insulin release due to activating the M3-subtype muscarinic receptors in the β-cell membrane [35]. On the other hand, the membrane-type receptor for bile acids (TGR5) was described as part of a mechanism related to β-cell regeneration and insulin release improvement [36]. Interestingly, PTs such as OA, betulinic acid (BA), and ursolic acid (UA) are potent and selective TGR5 agonists, and all of them have high structural similarity to SA, increasing the chance that molecules such as SA can act as TGR5 agonists [37].

A recent work reported numerous natural triterpenes that reduced the adverse metabolic effects of diabetes and related complications, including pancreatic β-cell damage [38,39,40]. For example, it was identified that 3β-hidroxihop-22(29)ene, obtained from *Croton heterodoxies Baillon*, stimulates glucose uptake and insulin secretion via the modulation of potassium and calcium channels in hyperglycemic rats [41]. Studies conducted in insulinoma cell lines and primary cultures of rat pancreatic islets exhibit increased insulin secretion via the increase in the Glut 2 expression when treated with compound K, a ginsenoside that has been widely studied for its antidiabetic effect [42].

Several studies identified the beneficial effect of triterpenes, which contribute to maintaining the integrity and function of pancreatic β-cells, as well as protecting them from the damage mediated by reactive oxygen species (ROS), reactive nitrogen species (RNS), and inflammation. Some of these compounds are boswellic acid [43], α and β amyrins [44], arjunolic acid [45], asiatic acid [23], calestrol [46], and ginsenosides [47,48]. Although there is relevant information regarding the mechanism of action of these compounds, studies validating their activity in other biological models, particularly their effect in humans, are required.

When assessing the effects of SA on lipid metabolism, a reduction in the total plasma cholesterol levels was identified. Considering that LDL and HDL levels do not change with respect to control animals fed with HFD, the decrease in total cholesterol levels may be associated with a decrease in triacylglycerol (TAG) levels (Figure 3A–D). The experimental induction of insulin resistance through HFD, stimulates hepatic TAG production and increases gluconeogenesis [49]. The present study showed a significant reduction in TAG levels, which may be associated with an inhibitory effect on the synthesis of hepatic TAG by SA.

There are various mechanisms potentially associated with TAG levels decreasing after SA exposition; the inhibition of human carboxylesterase type 1 (hCE1), an enzyme responsible of bulky alkyl chain ester such as TAGs, clearly related with obesity and T2DM progression [50]. Naturally occurring PTs has shown activity against hCE1 [51]. Interestingly, the carboxylic acid at C28 position, structural feature shared with SA, proved be the most important substitution in a 14 PTs series, granting activity and selectivity over human carboxylesterase type 2 (hCE2) [52]. TPs showed inhibitory activity against pancreatic lipase [53,54,55], its modulation reduced lipids absorption [56].

Recently, various compounds of vegetal origin were identified which in addition to their antihyperglycemic effect, contribute to the reduction of TAG levels. Compounds of triterpenoid origin have been reported for their antihypertriglyceridemic activity. Antcin K inhibits the fatty acid synthase, increases peroxisome α proliferator-activated receptor (PPAR-α) expression, and reduces the mRNA expression of sterol 1c regulatory element binding proteins (SREBP-1c) in the liver, thereby contributing to the decreased TAG levels and hepatic steatosis [57]. Similar results were obtained in the evaluation of eburicoic- [58] and dehydroeburicoic acids [59], two PTs obtained from *Antrodia camphorata*.

When evaluating adipokine levels in blood, it was observed that both metformin and SA significantly reduced leptin without affecting adiponectin levels (Figure 3E,F). Leptin functions as both a hormone and a cytokine. While functioning as the former, it regulates appetite and basal metabolism, whereas while functioning as the latter, it plays an important role in the hematopoiesis and angiogenesis, as well as in the innate and adaptive immune responses [60]. Increases in leptin secretion are associated with chronic inflammatory conditions [61]. States of insulin and leptin resistance coexist in obese individuals, and the association among these hormones, obesity, and the potential development of T2DM is evident [62].

The reduction of leptin levels in prediabetic mice treated with SA appears to be unrelated to a decrease in insulin resistance, as indicated by the homeostasis model assessment of insulin resistance (HOMA-IR) index. These results suggest that the effect in the reduction in adipocyte leptin synthesis is independent of the effect of this compound in β cells (insulin increase) and hepatocytes (decrease TAG synthesis). Recent studies have reported a decrease in leptin synthesis following the treatment of differentiated 3T3-L1 adipocytes, with a triterpene mixture [27]. It will be worthy to determine whether decreased leptin levels in animals treated with SA is owing to a local effect inhibiting leptin synthesis in adipocytes or whether the effect is systemic, where hormone sensitivity is increased. From experiments with triterpenes obtained from ginseng, Wu Y., 2018, reported that the ginsenoside Rb1 and certain saponins improve central leptin sensitivity, thereby reducing the levels of this hormone in the blood [63].

Considering that, macrophages are relevant cells in the secreting profile of the inflamed adipose tissue, the effect of SA in the expression of inflammation marker genes in the adipose tissue of diet-induced obese mice was evaluated. It was observed that in treated animals, the TNF-α, MCP-1, IL-6, and IL-1β expression is significantly reduced at a transcriptional level (see Figure 4A–D). These results match those obtained by Ceballos et al. 2018, who reported a significant reduction in the expression of proinflammatory genes in J774 mouse cells treated with a triterpene mixture [27]. Studies conducted by Patil et al., 2015, validated the effect of pentacyclic triterpenes in nuclear factor kappa B (NF-kB) activation, which is a canonical pathway for the activation of proinflammatory cytokines, such as TNF-α and IL-1β, which have an important role in the genetic expression of other proinflammatory cytokines [64].

The TNF-α and IL-6 play a key role on activation of the NF-κB pathway, the link between this process and insulin resistance is widely reported both in animal models and humans [65,66,67]. The NF-κB induces a negative regulation over insulin receptor substrate IRS1/2 [68], blocking the glucose transporter type 4 (GLUT4) translocation to membrane cellular increasing de insulin resistance and, consequently, contribute in a major way to hyperglycemic condition progression. This mechanism exemplified the characteristic snowball physiopathology of T2DM. Although the insulin sensibilization after treatment with SA has not been directly demonstrated, the significant reduction of proinflammatory gene expression in the adipose tissue of obese mice is a promising multitarget approach capable of slowdown of the aforementioned snowball.

On comparing the bodyweight of tested animals, no significant changes were identified in animals treated with metformin and SA when compared with control animals fed with HFD. Similar results were obtained when weighing the liver and the visceral adipose tissue (see Figure 5). With the idea of assessing whether lipid metabolism in adipocytes has any effect on weight conservation in the visceral adipose tissue; therefore, on body weight, the expression of encoding genes was evaluated for proteins involved in the cell catabolism (PPAR-α, PGC-1 and Cpt1A), and a protein in the synthesis of fatty acids (ACC). The results of gene expression in the adipose tissue of mice fed with HFD compared with mice on a regular diet are consistent with those reported in the literature [69].

Treatment with SA does not affect ACC, PPAR-α, and PGC-1 expressions when compared with control mice on an HFD (see Figure 4E–H). On the contrary, treatment with SA significantly reduced Cpt1a expression. These results suggest that, under the treatment approach used, SA does not affect the metabolism of visceral adipocyte lipids. Consequently, the weight of the visceral adipose tissue is not affected, and neither is body weight in general.

## 3. Materials and Methods

### 3.1. Instruments and Reagents

All high-performance liquid chromatography (HPLC) grade solvents used in the extraction protocol were obtained from Merck (Darmstadt, Germany). Ultrapure water was obtained using an Arium^®^-pro ultrapure system (Sartorius, Goettingen, Germany). Thin-layer chromatography was performed on Silica gel 60G F₂₅₄ 25 glass plates. Extract fractionation was performed with an Isolera™ One System from Biotage^®^ (Charlotte, North Carolina USA 28269). A precision ML/G3 Rotary Evaporator from Heidolph (Schwabach, Germany) was used for solvent concentration. The RapidVap vacuum evaporation system from Labconco (Kansas City, MO, USA) was used to assist the drying process. Ultrasonic bath Elmasonic E 120H (Singen, Germany) was utilized to dissolve samples, and accurate weighing was achieved with a Sartorius balance model MSE125P-100DU (Göttingen-Germany).

To identify the presence of SA in raw extracts before downstream processing, a MALDI-TOF/TOF analysis was performed using a Bruker Daltonics Ultraflextreme mass spectrometer (Billerica, MA, USA). The instrument is equipped with a 1 kHz Smart Beam Nd:YAG laser (355 nm) operated at 60% of the instrument arbitrary power scale (3.92 µJ/pulse). Negative ion mass spectra, from *m*/*z* 200 to 800, were acquired in reflectron mode with a pulsed ion extraction set at 100 ns and an accelerating voltage of 20 kV. Instrument calibration was performed using 1,5-diaminonaphthalene and a mixture of phthalocyanines, purchased from Sigma-Aldrich (St. Louis, MO, USA) covering the entire working mass range. Each reported analysis corresponds to the sum of 5000 mass spectra. Data analysis was performed using the Flex Analysis software version 3.4 (Bruker Daltonics, Billerica MA, USA). Experimental and theoretical isotopic patterns, calculated with ChemCalc [70], were compared to verify compound identification.

### 3.2. Plant Material Collection

*Cecropia telenitida* roots were collected in La Ceja, Antioquia, Colombia, in June 2018, at an altitude of 2454 masl and a geodesic location of 6°00′07.0″ N, 75°23′32.9″ W. Approximately 5 kg of roots was collected from three representative specimens with similar characteristics. Care was taken to not cause lethal injury to the sampled individuals. A taxonomist confirmed the identity of the collected specimens. A voucher (Alzate-Montoya 5189) is present in the herbarium of Universidad de Antioquia, Colombia. Proper authorization to collect wild species for noncommercial scientific research purposes was obtained from Colombian authorities (Resolution 0763 - 09 may of 2018. Program for the study, usage and sustainable exploitation of Colombian biodiversity).

### 3.3. Purification of SA from Cecropia telenitida Roots

The roots collected (5 kg) were dried at 45 °C for six days and later powdered using a disc sander and disposed at the 20 L percolator tank from a semi-pilot extraction plant. The vegetal material was extracted with 10 L of the following n-hexane, dichloromethane/ethyl acetate, ethyl acetate, ethyl acetate/methanol, and methanol solvents [10]. For every solvent system, the extraction process was maintained under constant temperature (40 °C) and continuous stirring rate (90 rpm) for 8 h.

For MALDI analysis, stock solutions of 1,8-bis(dimethylamino)-naphthalene (DMAN) were prepared in MeOH at 2 mM. A stock solution (2 mM) of the extract was prepared in MeOH. Equal volumes (2 µL) of matrix and extract stocks solutions were mixed using a vortex at 1000 rpm for 3 min. A 1.0 μL of each mixture was deposited on a ground steel MALDI plate and allow drying prior to mass spectrometry (MS) analysis. The extract with the major ion abundance at 499.342 *m*/*z* was n-hexane, as shown in Figure 1.

The empty cartridge (Biotage^®^ empty 340 g cartridge) were packed with silica (0.04 to 0.063 mm particle size), the proper packaging process were guaranteed assisted with pressurized air. This sample was fractioned using a gradient from 100% dichloromethane to 100% ethyl acetate during 15 column volumes using an automated flash chromatography system.

In order to avoid high molecular weight interferences, the samples containing SA were submitted to size exclusion chromatography by means of Sephadex LH-20. The purification process yielded 5 g SA (0.1%) with an optical purity of 98% by HPLC. The chemical characterization was carried out by routine mono- and bi-dimensional NMR experiments, and high-resolution mass spectrometry. Spectroscopic data were compared with our previous reports [12], and can be consulted in Appendix A.

### 3.4. Glucose and Insulin Tolerance Test in the Mouse Model of Insulin Resistance

C57BL6/J male mice (Charles Rivers Laboratories, Wilmington, MA, USA) over four weeks old were used in this study. Mice were housed at 22 ± 2 °C with a 12:12 h light-dark cycle with free access to food and water for eight weeks. Mice were randomly divided into four groups for this study. A control (Chow) group (*n* = 10) was fed a normal diet (ND, 14% fat/54% carbohydrates/32% protein). A high-fat diet (HFD) group (*n* = 30) was fed a high-fat diet (HFD, 42% fat/42% carbohydrates/15% proteins) and after 10 weeks under HFD, was divided into three different groups; HFD control group; HFD + (Metf.) group received 10 oral doses of metformin 200 µg administered by oral gavage during 10 days; HFD + (SA) group received 10 oral doses of 300 µg of SA by oral gavage during 10 days.

The Institutional Animal care and Use committee of the University of Antioquia (Protocol number 65) approved all animal studies. Before and after treatments, an intraperitoneal glucose tolerance test (IPGTT) was performed on the mice by administering a glucose load of 2.0 g/kg body weight. Insulin tolerance test (ITT) was performed after an overnight fast. Initial blood glucose levels were determined, followed by injection (ip) of human insulin 0.75 U/kg (Humulin, Eli Lilly, Indianapolis, IN, USA). Blood glucose levels were measured via tail vein blood at 0, 30, 60, and 120 min after the injection using a GlucoQuick Glucometer (Procaps, Barranquilla, Colombia). The zero time was measured just before glucose injection. After glucose metabolism studies, mice were sacrificed, and blood, liver, and adipose tissue were collected for further analysis.

### 3.5. Serum Biochemical Parameters Measurement

The following commercial kits were used to determine serum biochemical parameters: Total cholesterol (TC, 11,505), triacylglycerides (TAG, 11,528), high-density lipoprotein (HDL, 11,557), low-density lipoprotein cholesterol (LDL, 11,585) (BioSystems S.A, Barcelona, España); Leptin (ab100718) and Adiponectin (ab108785) (Abcam, Cambridge, UK). Insulin in mouse serum was quantified using an ultrasensitive enzyme immunoassay (10-1247-01 Mercodia, Uppsala, Sweden). All determination was carried out following the manufacturer´s recommendation and the absorbance of each sample was measured using a Varioskan™ LUX multimode microplate reader (Thermo-Fisher Scientific, Waltham, MA, USA).

### 3.6. RNA Extraction and Real-Time PCR

Total RNA was extracted from visceral white adipose tissue with the RNeasy kit (QIAGEN, Valencia, CA), and the reverse transcription reaction was performed with 500 ng total RNA, 50 ng/ul random hexamers, 10 mM dNTP Mix, 20 mM Tris-HCl pH 8.4, 50 nM KCl, 2.5 mM MgCl2, 40 U/ul RNase Out, and 200 U/ul Super Script III RT, (Invitrogen, MA, USA) according to the manufacturer’s instructions.

Real-time quantitative PCR (qPCR) analyses were performed with 50 ng cDNA and 100 nM sense and antisense primers (Integrated DNA Technologies, Coralville, IA, USA) in a final reaction volume of 25 μL by using the Maxima SYBR Green/ROX qPCR Master mix (Thermo Scientific, Waltham, MA, USA) and the CFX96 real-time PCR detection system (Bio-Rad, Hercules, CA, USA). Specific primers sequences are provided in Appendix A. Relative quantification of each gene was calculated after normalization to Cyclophilin RNA by using the comparative Ct method. The program for thermal cycling was 10 min at 95 °C, followed by 40 cycles of 15 s at 95 °C, 30 s at 60 °C, and 30 s at 72 °C.

### 3.7. Statistical Analysis

Results are expressed as means ± SEM. Comparisons between groups were analyzed using one-way analysis of variance (ANOVA) followed by a Dunnett post hoc test. The trapezoidal rule was used to determine the area under the curve (AUC). Homeostasis model assessment (HOMA) of insulin resistance (HOMA-IR) was calculated as (fasting glucose level x fasting insulin level/ 22.5 × 18), and HOMA of β-cell function (HOMA-%B) was calculated as (20 × fasting insulin level/ ((fasting glucose level/18)—3.5). Differences were considered as significant at *p* < 0.05. All analyses were performed with the Prism 8 (GraphPad software, Inc., La Jolla, CA, USA) statistical software.

## 4. Conclusions

SA is a good candidate for the development of a prototype drug for T2DM. This compound decreases glucose during fasting, increases insulin levels, and improves glucose and insulin tolerance. At the lipid metabolism level, SA reduces total cholesterol and triacylglycerol levels, confirming its beneficial effects for the treatment of T2DM, as well as for other metabolic complications associated with obesity, such as cardiovascular disease. On the other hand, obese animals treated with SA showed significantly reduced inflammation in the adipose tissues. These results stimulate further studies to identify the best treatment approach, improve compound solubility, and increase bioavailability.

## Figures and Tables

**Figure 1 molecules-25-01486-f001:**
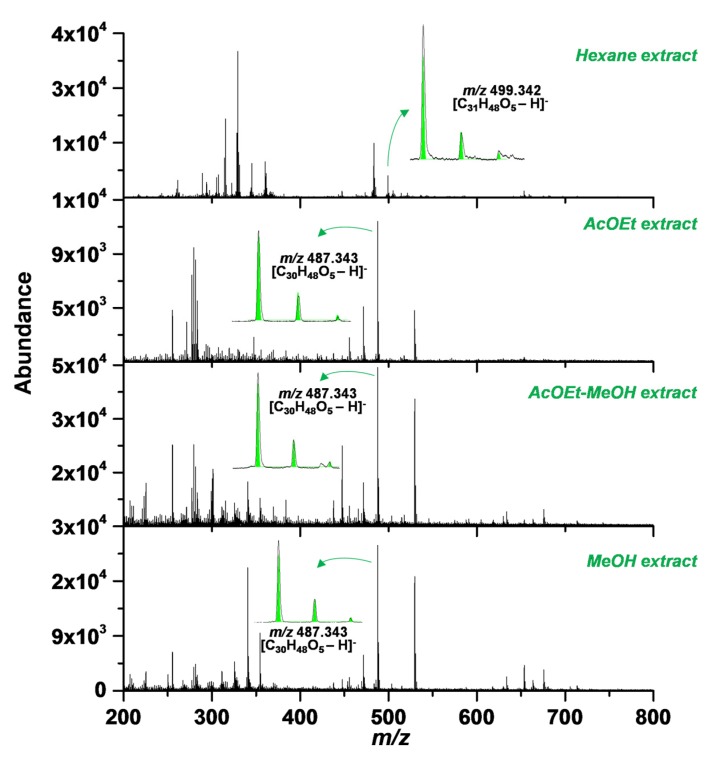
Matrix assisted laser desorption/ionization–time-of-flight (MALDI-TOF) detection of serjanic acid and isotopic ratio demonstrating its presence in the hexane extract. The theoretical exact mass (499.342 *m*/*z*) yields 2.00 ppm error.

**Figure 2 molecules-25-01486-f002:**
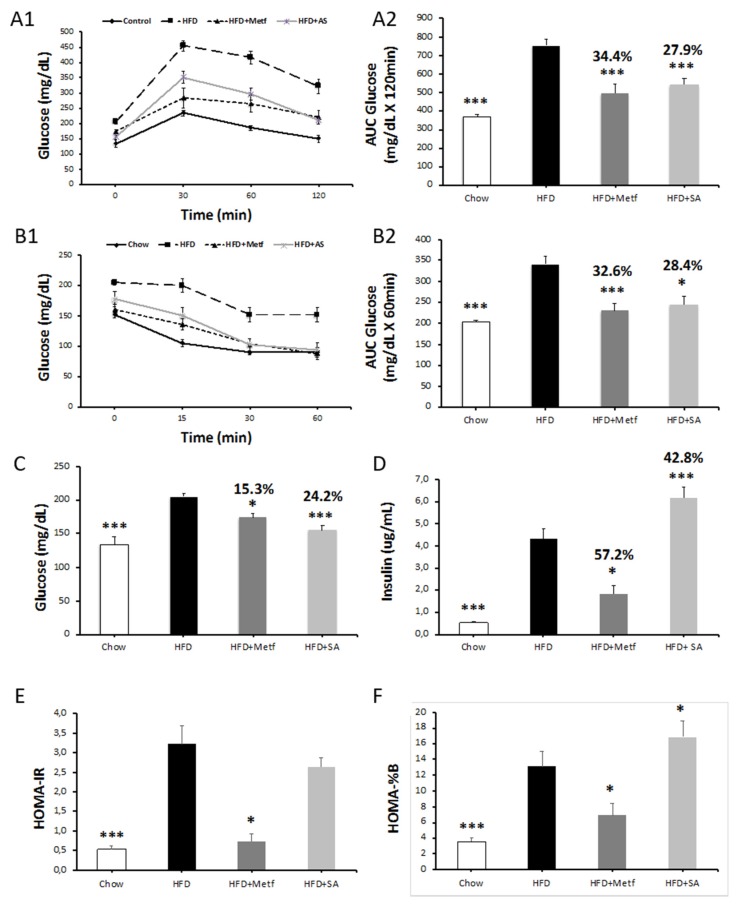
Assessment of carbohydrate metabolism and insulin secretion. Glucose tolerance test (IPGTT) (**A1**), insulin tolerance test (ITT) (**B1**), area under the curve (AUC) values (**A2**,**B2**), fasting blood glucose levels (**C**), fasting blood insulin levels (**D**), homeostasis model assessment of insulin resistance (HOMA-IR) (**E**), and homeostasis model assessment of β-cell function (HOMA-B) (**F**) in serjanic acid treated prediabetic mice. All assays were done after 10 treatment doses. The area under the curve (AUC) values were calculated using data obtained in A1 and B1. HFD: The percentages above the bars, represent the variation with respect to the high fat diet (HFD) group. High fat diet fed control group. HFD + mertformin (Metf.): High fat diet fed mice given 10 oral doses of metformin. HFD + SA: High fat diet fed mice given 10 oral doses of serjanic acid. (*n* = 7–9 per group) * *p* < 0.05 vs. HFD and *** *p* < 0.001 vs. HFD, analysis of variance (ANOVA) with Dunnett’s post hoc test; D and G Kruskal–Wallis test. Values are expressed as mean ± SEM.

**Figure 3 molecules-25-01486-f003:**
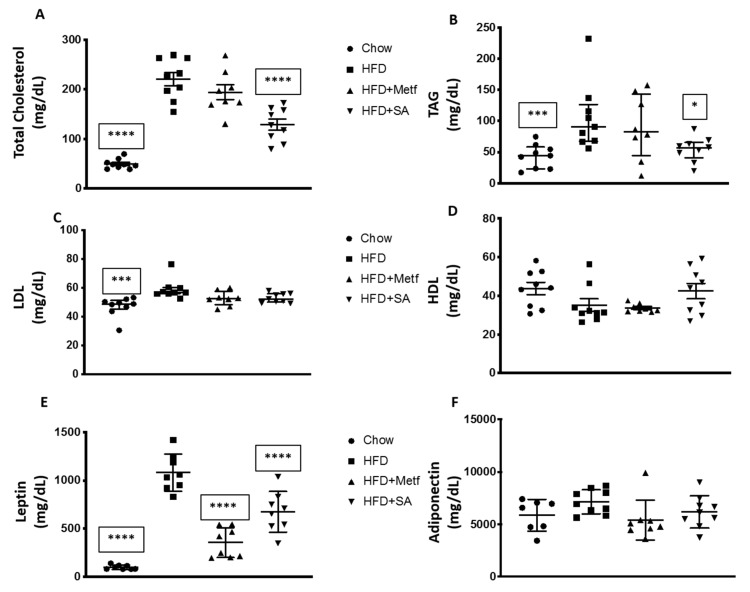
Effect of serjanic acid treatment in plasmatic parameters in a mouse model of diet-induced obesity. Total cholesterol (**A**), triglycerides (**B**), low density lipoprotein (LDL) cholesterol (**C**), high density lipoprotein (HDL) cholesterol (**D**), leptin (**E**), and adiponectin levels (**F**). HFD: High fat diet fed control group. HFD + Metf.: High fat diet fed mice given 10 oral doses of Metformin. HFD + SA: High fat diet fed mice given 10 oral doses of serjanic acid. * *p* < 0.05 vs. HFD, *** *p* < 0.001 vs. HFD, and **** *p* < 0.0001 vs. HFD, ANOVA with Dunnett’s post hoc test; C, D, and F Kruskal–Wallis test. Values are expressed as mean ± SEM and single data points retrieved from seven to nine animals.

**Figure 4 molecules-25-01486-f004:**
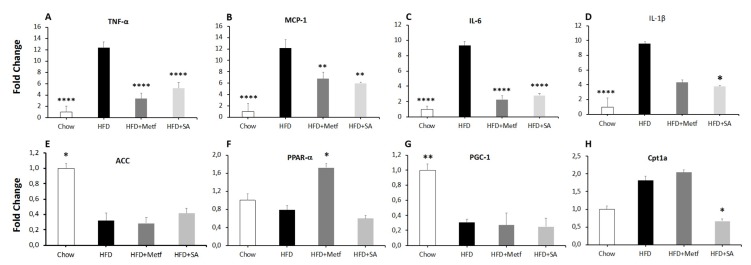
Effect of serjanic acid treatment on adipose tissue gene expression in a mouse model of diet-induced obesity. mRNA expression levels by quantitative reverse transcription -polymerase chain reaction (qRT-PCR) of tumor necrosis factor-alpha (TNF-α) (**A**), monocyte chemoattractant protein-1 (MCP-1) (**B**), interleukin-6 (IL-6) (**C**), interleukin-1 beta (IL-1β) (**D**), acetyl-CoA carboxylase (ACC) (**E**), peroxisomal proliferation-activated receptor-alpha (PPAR-α) (**F**), peroxisome proliferator-activated receptor gamma coactivator-1 (PGC-1) (**G**), and carnitine palmitoyltransferase 1-A (Cpt1A) (**H**) in adipose tissue of serjanic acid treated diabetic mice. HFD: High fat diet fed control group. HFD + Metf.: High fat diet fed mice given 10 oral doses of Metformin. HFD + SA: High fat diet fed mice given 10 oral doses of serjanic acid. (*n* = 4–5 per group) * *p* < 0.05 vs. HFD, ** *p* < 0.01 vs. HFD, and **** *p* < 0.0001 vs. HFD, ANOVA with Dunnett´s post hoc test; B and D Kruskal–Wallis test. Values are expressed as mean ± SEM.

**Figure 5 molecules-25-01486-f005:**
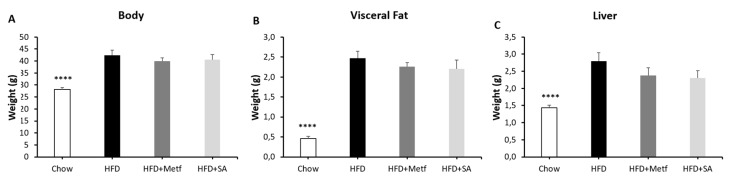
Effect of serjanic acid treatment on total body, visceral fat, and liver weights in a mouse model of diet-induced obesity. Whole body (**A**), visceral fat (**B**), and liver (**C**) weights were measured. HFD: High fat diet fed control group. HFD + Metf.: High fat diet fed mice given 10 oral doses of Metformin. HFD+SA: High fat diet fed mice given 10 oral doses of serjanic acid. (*n* = 7–9 per group) **** *p* < 0.0001 vs. HFD, ANOVA with Dunnett´s post hoc test. Values are expressed as mean ± SEM.

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
