# Peer review of "Serjanic Acid Improves Immunometabolic Markers in a Diet-Induced Obesity Mouse Model"

_molecules, 2020, doi:10.3390/molecules25071486_

Round 1

Reviewer 1 Report

Molecules Manuscript 741948. Serjanic acid improves immunometabolic markers in a diet-induced obesity mouse model.

L 41-65. The authors should proofread and check their English grammar and word use throughout the manuscript, but especially in the introduction. Several words are misused, such as “endemic,” “development of fatalities,” “low cost treatments alike phytotherapeutics,” and L 85 “unluckily”

L 48-57. The overview of the relationship of obesity to insulin resistance in this paragraph is quite general and somewhat vague. The authors suggest that overeating and high caloric intake causes hypertrophy in adipocytes, which are then altered, and eventually generates insulin resistance. While not necessarily wrong, these statements are far too general. For example, visceral adipose is more of a factor in IR than subcutaneous adipose. Authors mention “oxidative stress” and “inflammation” instead of more specific factors, such as inflammatory cytokines. The authors need to set up the framework in the introduction to establish a connection between the mechanism of insulin resistance and a potential mechanism by which serjanic acid might be a beneficial treatment/supplement. This is unclear in the current manuscript.

L 58-65. While the authors make a point about the economic burden of treatment for diabetes, it is unknown if a treatment based on serjanic acid will be low-cost. This paragraph does not add helpful content to the whole of the manuscript and should be deleted.

L 75-85. Similar to the earlier paragraph (48-57) this paragraph is far too vague to be helpful to the reader. The authors include vague statements such as, “prevention of a variety of human diseases,” “exhibit immunomodulatory activity as they inhibit one or several pathways involved,”  “reported to have a variety of biological effects, but inflammation and hypoglycemic seem to be consistent.”  The authors must establish a clear relationship and/or potential mechanism for serjanic acid as a treatment for IR.

L 83-85. I do not understand the last sentence of this paragraph. I do not understand enough of what the authors are trying to say to even suggest how to rewrite. Was SA identified by extracting mixtures, or is it a part of mixtures within plants, and what is meant by, “without details of its content”? The content of what? The mixtures? Or unidentified compounds in the plant? Or previous studies with proprietary undisclosed mixtures? I’m sorry, I don’t understand this sentence at all.

Figure 2. The “Chow” label is not described in the label, or elsewhere in the manuscript. I understand “chow” is a common feed, so I assume these are the control, normal diet mice? It should also be noted in the label whether these mice are non-obese controls eating a normal diet or if they are instead obese controls eating a normal diet.

Figure 2 Labels. The A-H labeling is not intuitive for the reader, especially for A-to-D graphs. The Figure 2 label worsens this by describing in order, A, C, E, F, G, H, followed by B and C, almost as an afterthought. Suggest re-labeling A & B as A1, A2, and then define in the Figure label that A1 is the glucose tolerance curve and A2 is the AUC of that curve. Similarly, label C & D as B1, B2, (B1 is insulin tolerance curve and B2 is the AUC). Then adjust the remainder of the letters to D, E, F, G and describe these in the label after A1, A2 and B1, B2.

Figure 2 Significance designations. The authors compare Chow to the High Fat Diet (HFD) and the Metformin and SA treated mice to the HFD. But there is no comparison of the treated mice to the Chow/control mice. It would be more beneficial to compare all bars, treatments to HFD and the Controls. The reason is that it is difficult to compare a “significant” effect to a biologically relevant effect. For example, in graph B, the Chow, HFD+Metf and HFD+SA are all different from the HFD-non treated mice. But are the HFD+Metf and HFD+SA similar or different from the Chow/Control (I assume non-obese, but that’s another comment) mice? This would be biologically relevant. All graphs should be altered to clarify this point. Suggest using a, b, c, notations rather than * to designate differences between bars.

L 136-137. “The increasing of insulin ensuing SA administration…” This sentence does not appropriately describe the results. Insulin was not shown to “increase” (on a slope or curve) as a direct result of SA. Fasting blood insulin was higher after 10 treatment doses of SA over 10 days.

Also, during the discussion of this point, the authors speculate why fasting insulin was higher in SA treated animals, as if this is a benefit. Perhaps I am missing something, but in an insulin-resistant model, higher fasting insulin is not a benefit. Graphs C and D show improvement in insulin tolerance in SA-treated mice, but it is unknown if they are improved enough to be similar to Chow/controls. Graph G indicates no difference in insulin resistance in SA vs HFD mice, which would create higher fasting insulin, which is not a benefit. Of course, Graph G (as well as H) are both derived from and rely heavily upon the data in Graphs E and F rather than dynamic tests of insulin dysregulation or beta cell function. Thus, the authors should use caution in their speculation whether higher fasting insulin is a benefit of SA treatment. This should be noted both in the results and in L 358 of the conclusions.

Figure 5. “Weight” on the Y-axis is misspelled/typo.

L 308-313. Were all mice non-obese at the start, then fed either the normal diet (control mice remained non-obese) or the HFD (which induced obesity)? Or were all mice initially obese and then fed a normal vs HFD? The methods must support that this is an obesity model and not a diet model.

L 349-355. Statistical analysis. Were the data all tested for normality? Personal experience with similar data suggest that some may not fit a normal distribution and require transformation before parametric tests may be applied. This is not always the case, but authors should verify that data were tested for normality and fit a normal distribution for parametric testing.

Author Response

Response to Comments from Reviewer 1

L 41-65. The authors should proofread and check their English grammar and word use throughout the manuscript, but especially in the introduction. Several words are misused, such as “endemic,” “development of fatalities,” “low cost treatments alike phytotherapeutics,” and L 85 “unluckily”

Response 1:

L 41: “Type 2 diabetes mellitus (T2DM) has reached endemic proportions” has been replaced with “Type 2 diabetes mellitus (T2DM) is a growing health issue worldwide”.

L 48: “development of fatalities” has been replaced with “development of a number of conditions”.

L 64: “low cost treatments alike phytotherapeutics” has been deleted according the reviewer recommendation

L 91: “but unluckily without details of its content” has been replaced with “without a detailed description of its potential properties”.

L 48-57. The overview of the relationship of obesity to insulin resistance in this paragraph is quite general and somewhat vague. The authors suggest that overeating and high caloric intake causes hypertrophy in adipocytes, which are then altered, and eventually generates insulin resistance. While not necessarily wrong, these statements are far too general. For example, visceral adipose is more of a factor in IR than subcutaneous adipose. Authors mention “oxidative stress” and “inflammation” instead of more specific factors, such as inflammatory cytokines. The authors need to set up the framework in the introduction to establish a connection between the mechanism of insulin resistance and a potential mechanism by which serjanic acid might be a beneficial treatment/supplement. This is unclear in the current manuscript.

Response 2:

L 52-53: “the endocrine function of the visceral adipose tissue is altered. Hypertrophic adipocytes secrete several adipokines that are directly associated with inflammation and a microenvironment is established” has been added according the reviewer recommendation.

L 58-65. While the authors make a point about the economic burden of treatment for diabetes, it is unknown if a treatment based on serjanic acid will be low-cost. This paragraph does not add helpful content to the whole of the manuscript and should be deleted.

Response 3:

L 59-66, 388-393: Paragraph and references have been deleted according the reviewer recommendation.

L 75-85. Similar to the earlier paragraph (48-57) this paragraph is far too vague to be helpful to the reader. The authors include vague statements such as, “prevention of a variety of human diseases,” “exhibit immunomodulatory activity as they inhibit one or several pathways involved,” “reported to have a variety of biological effects, but inflammation and hypoglycemic seem to be consistent.”  The authors must establish a clear relationship and/or potential mechanism for serjanic acid as a treatment for IR.

Response 4:

L 76-98: Paragraphs have been reorganized in order to clarify concepts and establish a more clear potential mechanism for serjanic acid as a treatment for IR.

L 83-85. I do not understand the last sentence of this paragraph. I do not understand enough of what the authors are trying to say to even suggest how to rewrite. Was SA identified by extracting mixtures, or is it a part of mixtures within plants, and what is meant by, “without details of its content”? The content of what? The mixtures? Or unidentified compounds in the plant? Or previous studies with proprietary undisclosed mixtures? I’m sorry, I don’t understand this sentence at all.

Response 5:

L 83-85: Paragraph has been rewritten L 90-92.

Figure 2. The “Chow” label is not described in the label, or elsewhere in the manuscript. I understand “chow” is a common feed, so I assume these are the control, normal diet mice? It should also be noted in the label whether these mice are non-obese controls eating a normal diet or if they are instead obese controls eating a normal diet.

Response 6:

L 320: Chow group label has been included in the control group description.

Figure 2 Labels. The A-H labeling is not intuitive for the reader, especially for A-to-D graphs. The Figure 2 label worsens this by describing in order, A, C, E, F, G, H, followed by B and C, almost as an afterthought. Suggest re-labeling A & B as A1, A2, and then define in the Figure label that A1 is the glucose tolerance curve and A2 is the AUC of that curve. Similarly, label C & D as B1, B2, (B1 is insulin tolerance curve and B2 is the AUC). Then adjust the remainder of the letters to D, E, F, G and describe these in the label after A1, A2 and B1, B2.

Response 7:

L 133-143: Figure 2 labels have been reorganized.

Figure 2 Significance designations. The authors compare Chow to the High Fat Diet (HFD) and the Metformin and SA treated mice to the HFD. But there is no comparison of the treated mice to the Chow/control mice. It would be more beneficial to compare all bars, treatments to HFD and the Controls. The reason is that it is difficult to compare a “significant” effect to a biologically relevant effect. For example, in graph B, the Chow, HFD+Metf and HFD+SA are all different from the HFD-non treated mice. But are the HFD+Metf and HFD+SA similar or different from the Chow/Control (I assume non-obese, but that’s another comment) mice? This would be biologically relevant. All graphs should be altered to clarify this point. Suggest using a, b, c, notations rather than * to designate differences between bars.

Response 8:

Figure 2: We understand the reviewer comment, the presence of the Chow group is to confirm metabolic changes when fed mice HFD but the study groups (HDF+Metf and HDF+SA) have the same fat diet as HDF group, so we consider we have to compare them with HDF group, since Chow group have a different type of diet.

L 136-137. “The increasing of insulin ensuing SA administration…” This sentence does not appropriately describe the results. Insulin was not shown to “increase” (on a slope or curve) as a direct result of SA. Fasting blood insulin was higher after 10 treatment doses of SA over 10 days.

Response 9:

L 144-145: “The increasing of insulin ensuing SA administration…” has been replaced with “Fasting blood insulin was increased after 10 treatment doses of SA”.

Also, during the discussion of this point, the authors speculate why fasting insulin was higher in SA treated animals, as if this is a benefit. Perhaps I am missing something, but in an insulin-resistant model, higher fasting insulin is not a benefit. Graphs C and D show improvement in insulin tolerance in SA-treated mice, but it is unknown if they are improved enough to be similar to Chow/controls. Graph G indicates no difference in insulin resistance in SA vs HFD mice, which would create higher fasting insulin, which is not a benefit. Of course, Graph G (as well as H) are both derived from and rely heavily upon the data in Graphs E and F rather than dynamic tests of insulin dysregulation or beta cell function. Thus, the authors should use caution in their speculation whether higher fasting insulin is a benefit of SA treatment. This should be noted both in the results and in L 358 of the conclusions.

Response 10:

Type 2 diabetes mellitus is a disease with reduced both insulin secretion and insulin sensitivity. The mouse model of diet-induced obesity (HDF group) used is a pre-diabetic mouse model, where there is an increase of insulin production, as observed in figure 2F, trying to compensate the insulin resistance generated by fat diet. SA increased significantly the production of insulin compared to the HDF group that may indicate that SA is stimulating pancreatic β cells activity, this could be a benefit for diabetic patients with a decrease insulin production.

Figure 5. “Weight” on the Y-axis is misspelled/typo.

Response 11:

Figure 5: “Weight” has been replaced with “Weight”.

L 308-313. Were all mice non-obese at the start, then fed either the normal diet (control mice remained non-obese) or the HFD (which induced obesity)? Or were all mice initially obese and then fed a normal vs HFD? The methods must support that this is an obesity model and not a diet model.

Response 12:

L 303-310: All mice were non-obese at the initial point, then fed either the normal diet or the HFD that induced obesity for 10 weeks. It is a mouse model of diet-induced obesity widely described.

Wang CY1, Liao JK. A mouse model of diet-induced obesity and insulin resistance.  Methods Mol Biol. 2012;821:421-33. doi: 10.1007/978-1-61779-430-8_27.

L 349-355. Statistical analysis. Were the data all tested for normality? Personal experience with similar data suggest that some may not fit a normal distribution and require transformation before parametric tests may be applied. This is not always the case, but authors should verify that data were tested for normality and fit a normal distribution for parametric testing.

Response 13:

Statistical analysis: All the data were tested for normality, for those which did not pass we used Kruskal-Wallis test as indicated in figure 3.

Reviewer 2 Report

Journal: Molecules
Manuscript ID: molecules-741948
Type of manuscript: Article
Title: Serjanic acid improves immunometabolic markers in a diet-induced obesity mouse model 
Authors: Gustavo Gutiérrez, Deisy Giraldo-Dávila, Marianny Y. Combariza, Ulrike Holzgrabe, Jorge Humberto Tabares-Guevara, José  R. Ramírez-Pineda, Sergio Acín, Diana Lorena Muñoz, Guillermo Montoya, Norman Balcazar

Based on a semipilot scale extraction established, serjanic acid was purified from the roots of Cecropia telenitida using direct MALDI-TOF analysis and evaluated in a diet induced obesity mouse model against several selected immunometabolic markers. The results showed that serjanic acid decreased fasting glucose, increased insulin levels, improved glucose tolerance and insulin sensitivity, and reduced total cholesterol and triacylglycerol levels and inflammation in the adipose tissue. This manuscript reports potential anti-diabetic activity of triterpene serjanic acid and provides an important reference for the design anti-diabetic agent based on triterpenes. Thus, it is recommended to be published as a potential Article in Molecules after minor changes shown in the attached pdf manuscript file.

Author Response

Reviewer 2:

Response 1:

All spelling changes were considered